# LGAD-Based Silicon Sensors for 4D Detectors

**DOI:** 10.3390/s23042132

**Published:** 2023-02-14

**Authors:** Gabriele Giacomini

**Affiliations:** Instrumentation Division 535B, Brookhaven National Laboratory, Upton, NY 11973-5000, USA; giacomini@bnl.gov; Tel.: +1-631-344-4825

**Keywords:** LGAD, avalanche photodiodes, APD, silicon sensors, timing, impact ionization, gain, spatial resolution

## Abstract

Low-Gain Avalanche Diodes (LGAD) are a class of silicon sensors developed for the fast detection of Minimum Ionizing Particles (MIPs). The development was motivated by the need of resolving piled-up tracks of charged particles emerging from several vertexes originating from the same bunch-crossing in High-Energy Physics (HEP) collider experiments, which, however, are separated not only in space but also in time by a few tens of picoseconds. Built on thin silicon substrates and featuring an internal moderate gain, they provide fast signals for excellent timing performance, which are therefore useful to distinguish the different tracks. Unfortunately, this comes at the price of poor spatial resolution. To overcome this limitation, other families of LGAD-based silicon sensors which can deliver in the same substrate both excellent timing and spatial information are under development. Such devices are, to name a few, capacitively coupled LGADs (AC-LGAD), deep-junction LGADs (DJ-LGAD) and trench-isolated LGADs (TI-LGADs). These devices can be fabricated by even small-scale research-focused clean rooms for faster development within the scientific community. However, to scale up production, efforts towards integrating these sensor concepts in CMOS substrates, with the obvious advantage of the possibility of integrating part of the read-out electronics in the same substrate, have begun.

## 1. Introduction

In high-energy physics (HEP) experiments, minimum ionizing particles (MIPs) emerge from vertexes, and their tracks are reconstructed by recording hits in the detectors surrounding the interaction point. At small radii, these detectors are made of silicon (pixels and strips) sensors. An MIP releases a signal of 80 electrons/hole pairs in silicon, a fair signal which is easily detected by modern read-out electronics. In fact, HEP and nuclear physics (NP) experiments started using silicon detectors back in 1980 [1]. If thinner sensors are used, either to keep the applied voltage reasonably low after irradiation or to limit the Coulomb scattering, the signal is proportionally reduced according to the thickness, and the noise increases due to the increase in the capacitance. Reading-out at shorter shaping time to have good timing is therefore challenging, as the noise is proportional to C/τ, and delivering good timing is not possible. Timing is not an issue when the multiplicity of hits is low enough and the tracks can be uniquely resolved. For example, the ATLAS [2] and CMS [3] experiments at CERN count millions of channels in their innermost layers which are equipped with thin silicon sensors. They are read-out by relatively slow front-end read-out electronics based on charge sensitive amplifiers, and signals are typically a few hundreds of nanoseconds after shaping. However, in the future upgrades, an increase in luminosity will create several vertexes for the same bunch crossing, and assigning hits to the correct track will be difficult. This scenario will be mitigated if we can assign a time information to the track and its associated hits in the order of a few tens of picoseconds. This value is deemed appropriate as same bunch-crossing vertexes are separated in time by this time interval. Assigning a timing information will allow the easing of the reconstruction, disentangling multiple primary and secondary vertexes and reducing ghost tracks.

To specifically address this scenario, a new silicon detector has been developed, as the older generation of silicon sensors and their read-out electronics cannot cope with the new timing requirements. Pioneered by the Centro Nacional de Microelectronica (CNM, Barcelona, Spain) [4] under the patronage of the CERN-RD50 community, the low-gain avalanche diode (LGAD) has been developed [5,6]. It is a silicon sensor that can deliver fast pulses with the required timing resolution because it is fabricated on thin substrates, and the signals are internally amplified.

The former feature allows carriers generated by the MIP to rapidly drift to the electrodes in just below one nanosecond for a typical LGAD thickness of 50 μm. The reduced thickness brings low signals and high capacitance; therefore, the signals need to undergo multiplication by a factor of a few tens which more than compensates for the reduced thickness. A signal of 80 k electrons can be fed to the read out electronics in the typical case of 50 μm thick active substrate and a gain of 20. However, higher gains are easily achievable. As compared to Avalanche PhotoDiode (APD), having maximum gains of a few hundreds, the gain is kept low in the order of a few tens as the signal generated by an MIP is large enough for the application.

Of course, the sensor alone is not enough, and a dedicated fast electronics is required. A transimpedance or RF amplifier needs to replace the slower charge sensitive amplifier in the read-out. The bandwidth increases (GHz) and so does the noise, but the signal–to–noise ratio (S/N) is kept to a good enough value by the amplified signals.

Currently, a few research facilities around the world are involved in the development of LGADs, such as Brookhaven National Laboratory (BNL, Upton, NY, USA) [7], CNM, Fondazione Bruno Kessler (FBK, Trento, Italy) [8], Hamamatsu (Japan) [9], IHEP-NDL (Beijng, China) [10] and Micron (Lancing, UK) [11]. In addition, read-out electronics are actively being developed. For evaluation of LGADs, dedicated amplifier boards have been developed as well and proficiently used: the University of California at Santa Cruz developed a single channel transimpedance amplifier [12] and Fermi National Accelerator Laboratory a 16-channel RF amplifier [13]. Microelectronics groups around the world are engaged in the design of several fast, low-power Application Specific Integrated Circuits (ASICs) for the read-out of LGADs.

Fast signals read out by fast electronics lead to the sought-after timing of a few tens of picoseconds, fast enough to resolve the tracks emerging from the same bunch crossing. However, spatial resolution in LGADs is very poor because of the lack of multiplication at the pixel or strip border. This no-gain region acts as a dead area, as the signal will most likely stay below the threshold for detection of the read-out electronics. A basic electrostatic simulation can show how a dead border [14] several microns large is an unavoidable feature of this sensor. To have a decent fill factor, the size of a pixel must therefore be very large compared to the thickness of the substrate. For example, LGAD pixel arrays for the CMS and ATLAS timing detectors have a pitch of 1.3 mm [15,16]. Strips or pixels at a pitch of a few hundreds of microns are excluded [17].

A new line of solid-state devices was initiated to overcome this severe limitation. The starting point is the standard LGAD, with its good timing performance, and other devices based on it have been conceived and fabricated. In this paper, we present the basic structure of an LGAD and describe the other silicon sensors which are based on it which also show good spatial resolution. The work towards a 4D detector is quite active and is already transferring these concepts to CMOS technologies, which bring obvious advantages as demonstrated by the deployment of monolithic active pixel sensors (MAPS) in the inner barrels of HEP and NP experiments.

## 2. LGAD

The basic unit of an LGAD is a p–n junction, where a shallow phosphorus layer is implanted on a thin p-type substrate (Figure 1). The n is implanted at a high dose to have low sheet and contact resistances. It is contacted by a metal layer (usually aluminum as it is typical in all other silicon sensors) and then connected to the read-out electronics by wire (or bump) bonding. As in any silicon sensor where n electrodes are patterned, it is necessary to isolate them from each other because the fixed positive oxide charge induces an electron layer at the oxide/silicon interface which would short-circuit the n pixels. The standard solution, valid also for LGADs, is to introduce a boron layer compensating for the electron accumulation layer between the pixels by either patterning a boron layer surrounding the n-electrodes (p-stop) or by the use of a uniform layer (p-spray) over the entire surface [14].

As they are engineered to detect MIPs traversing the substrate with fast timing, a high electric field which is traversed by electrons is required. In fact, as ionization coefficients in silicon are higher for electrons than for holes (for the same electric field), the gain is due to the multiplication of electrons. High-enough electric fields are created by implanting a deep boron layer a fraction of a micron to a few microns under the n${}^{+}$ phosphorus already implanted at the interface. Upon depletion of the boron layer, a high electric field develops. LGADs do not work in Geiger mode but rather in the linear region. Here, only electrons are multiplied as electric fields are low enough for a significant hole multiplication to occur, preventing the onset of avalanche breakdowns. While the electrons are multiplied and create secondary electrons and holes, these secondary holes, which drift back towards the p${}^{+}$ contact, are responsible for the majority of the signal, according to Ramo’s theorem [18].

As mentioned in the Introduction, these active layers are necessary to achieve fast timing, as the signal must develop in a very short time (order of one nanosecond). Typically, 20–50 μm thick substrates are used for LGADs. Timing detectors used in the upgrades of the ATLAS and CMS experiments plan to use 50 μm thick LGADs. A standard clean-room (but the same applies to CMOS foundries) cannot process such thin substrates unless they are placed on a thick carrier wafer (order of a few hundreds of microns thick). Initial substrates are of two possible kinds: a thin epitaxial layer grown over a thick low-resistivity handling wafer or a high resistivity wafer wafer-bonded to a low resistivity wafer and then thinned down to the desired thickness (by wafer grinding followed by a polishing step, chemical mechanical polishing (CMP) or dry polishing). In both cases, the function of the low-resistivity carrier wafer is dual: it acts as mechanical support for all the procedures in the clean room and as an ohmic contact (and a sink for the leakage and signal holes). Notice that LGADs are intrinsically single-sided devices, as the support wafer is unpatterned.

At the border of the pixel, defined by the shallow n${}^{+}$, an additional implant is placed. It is a deep, low dose, phosphorus implant, and therefore it is ohmically connected to it. It is called the junction termination edge (JTE) [19]. Its purpose is to reduce the electric field value at the edges of the shallow n${}^{+}$ so that the maximum electric fields develop within the gain layer. This solution is found in any CMOS APD as well [20], although it may have different shapes. The p-stop, if present, is implanted at a short distance from the JTE to limit the extension of the dead region, while keeping the distance large enough to prevent premature breakdown.

The gain layer, which is found under the n${}^{+}$, is usually implanted for better control of its doping profile. The extension of the gain layer must be within the border of the n${}^{+}$ (or the JTE) to avoid premature breakdown. Implantation energy and dose are extremely critical parameters and therefore must be constant over all the implantation region to guarantee the same multiplication value within the gain area. The implantation energy is in the range of 400 keV to 2 MeV, giving doping profiles about just below 1 μm to 2 μm deep. The multiplication value depends on the impact ionization integral, which is a complicated function of the electric field. A deeper gain layer requires lower electric fields (which are in turn generated by a lower implantation dose). In addition, since the ratio of the impact ionization coefficients for electrons and holes is a function of the electric field, lower electric fields give higher ratios resulting in less multiplication noise. Implantation doses of the order of 2 × $10^{12}$ cm${}^{-2}$ are typical. Pictures of the front side of LGAD devices fabricated at BNL are shown in Figure 2.

### 2.1. Timing in LGADs

In Figure 3, the software Weightfield2 [21] has been used to simulate current signals at the electrodes after an MIP crosses vertically either a diode or an LGAD. In these simulations, as the substrate thickness is much smaller (50 μm) than the lateral dimension (300 μm), devices can be thought of as one-dimensional.

Both devices initially have a null rise time because the signal starts developing as soon as the carriers (electrons and holes) move into the bulk. However, for later times, the signal decreases in the diode, while it increases in the LGAD because of the contribution of the multiplied holes drifting to the back contact in the LGAD. In both cases, the full signal develops in about one nanosecond. To have signals completely developed in this time frame, the electric field in the substrate should be high enough for the holes to rapidly drift and be collected at the back contact. An electric field of about 3–4 V/μm is enough for the multiplied holes to drift all the way through the substrate thickness in less than one nanosecond for a 50 μm thick LGAD.

Furthermore, any read-out electronics have a limited bandwidth, which will cause a non-null rise time for both devices and also a spreading of both signals beyond the nanosecond.

The timing resolution of LGADs takes contributions from a few terms, which add in quadrature: the jitter, the time walk, the Landau noise and the time-to-digital converter (TDC) binning. The jitter ($\sigma_{n}/dS/dt$) can be minimized with proper low-noise $\sigma_{n}$ read-out electronics: the slew rate (dS/dt) can be increased using large signals *S* (i.e. large gains) and thin substrates biased at voltages high-enough to achieve carrier velocity saturation. The time-walk noise contribution is due to the statistical distribution of the signal amplitudes. Signals vary because of two reasons: the Landau distribution of the charge generated by the MIPs and the fluctuations of the gain. The time of arrival, as detected by a trigger, depends on when the signal exceeds a threshold which is dependent on its amplitude. By correcting the amplitude through normalization (for example by means of the constant fraction discrimination method), this noise term can be made negligible.

The noise associated to the digitization error (TDC binning) can be made negligible as well.

The main term contributing to the timing noise is due to an intrinsic property of silicon. The Landau noise is due to the non-uniform generation of charge along the track of the MIP, which induces a variation of the signal shape. There are sensors that do not suffer from such noise; 3D sensors with their electrodes perpendicular to the surface are insensitive to this kind of noise, but only in the case of MIP crossing the device parallel to the columns. Thin substrates exhibit a lower Landau noise and a timing resolution of less than 30 ps has been measured on 50 μm thick devices, which goes down to about 15 ps in 20 μm thick devices. These are quite impressive timing resolutions, and any 4D detector needs to feature at least the level of this timing performance.

### 2.2. Spatial Resolution in LGAD

A numerical TCAD simulation of the electrostatic inside the bulk of an LGAD will make apparent how far away from the border region, as well as how far below the n${}^{+}$ and gain layers, the electric field lines are vertical and parallel to each other. Electrons generated here drift to the n${}^{+}$ and get multiplied as expected. However, close to the JTE and even below the gain implant, electric field lines point toward the JTE and away from the center of the device. If MIPs cross in this region, the signal electrons are collected by the JTE where only low electric fields exist and therefore are not amplified. For MIPs passing through this region, an LGAD behaves as a diode (gain = 1), and the signals will most likely be under the threshold for detection with fast electronics. In addition, electrons generated in the gap region between two JTEs, belonging to two neighboring pixels, are collected by these JTEs and are not amplified. The gap itself acts as a dead region for MIP detection. The extension of this dead region can easily be in the order of a few tens of microns. Fine pitch strip and pixels, with pitches in the order of 100 μm, are therefore excluded in the LGAD basic technology as the dead areas will lower the fill factor considerably. For example, both CMS and ATLAS use LGAD arrays having a pitch of 1.3 mm between pixels. One can certainly mitigate the problem by using some design or system features. Some techniques that have been found useful in increasing the fill factor consist of utilizing a narrow JTE with a gain layer ending close to the JTE or using a p-spray instead of a p-stop. In addition, tuning the gain layer to operate the LGAD at high bias voltages helps in straightening the electric field lines at the pixel border to effectively increase the active area. However, these tricks do not completely solve the problem, and this motivated the research towards silicon sensors which retain the good timing of the LGAD but also feature good spatial resolution.

## 3. Trench-Isolated LGAD

In trench-isolated LGADs (TI-LGADs) [22], fine pitch n-electrodes are implanted close together and isolated by a trench. The approximately 1 μm wide trench replaces all the structures, such as p-stop and JTE, in the border region of a standard LGAD. Pixels, separated by trenches, are very close together which usually comes at the price of an increased interpixel capacitance. In TI-LGADs, this issue is mitigated by two factors: trenches, while electrically separating the pixel, also keep the interpixel capacitance lower than the case of the same geometry and different isolation (p-spray or p-stop). Second, interpixel capacitance becomes less important than the capacitance towards the back for thin substrates. This point applies also to standard LGADs. In TI-LGADs, JTE and p-stop are no longer needed, which benefits the fill factor. The doping profile of the shallow n${}^{+}$ phosphorus implant and of the p${}^{+}$ boron gain layer are unchanged with respect to the standard LGAD so that the breakdown voltage (and related operational point and gain factor) stays the same in the two cases. The devices have been fabricated by Fondazione Bruno Kessler (FBK, Trento, Italy), leveraging their experience in silicon etching and LGAD fabrications, and currently they are the only producer. Silicon etching is used by FBK for 3D detectors, where columns are vertically etched into the substrate and doped by diffusion in furnaces. Aspect ratio (depth to diameter of columns) can be about 20:1; therefore, as the substrate thickness is about 100–150 μm, column diameters are in the order of 10 μm or just below. Trenches are used also in SiPMs [23] to optically isolate the single photon avalanche diodes (SPADs) so that the photons induced in one avalanche do not trigger parasitic avalanches in nearby SPADs (cross talk). To serve this purpose, trenches need to be filled with metal; a depth of only a few microns is required, which makes these trenches much more shallow than the columns in a 3D sensor. In TI-LGAD, trenches are quite similar to the SiPM ones, except that they are filled with oxide. At this time, details of the geometry have not been made public. A 7 μm dead border, instead of the usual few tens of microns, has been measured on these devices for an outstanding 75% fill factor for a 50 μm pitch sensor. Current–voltage characteristics are identical to those measured in standard LGAD devices with the same doping profiles. However, the current generation of TI-LGADs do not completely solve the problem as they do not yet achieve 100% fill factor which means further optimization may still be possible. Other LGAD-based devices can achieve 100% fill factor, as detailed below.

## 4. Inverted LGAD

An attempt to obtain high spatial resolution with a structure based on LGADs is the inverted LGAD (i-LGAD) [24]. It is based on keeping the n${}^{+}$ and gain layer on one side of the wafer, while the hole collecting electrodes are on the opposite side of the wafer [13,25] (a sketch is shown in Figure 4). The n${}^{+}$ and the gain layer will be uniformly implanted over a large area, interrupted only at the edge to allocate the proper termination for high voltage handling. The uniformity of the gain layer over a large area assures a 100% fill factor. On the opposite side, the hole collecting electrodes can be placed at a small pitch without detrimental effects on the uniform multiplication occurring at the opposite side of the wafer. A good spatial resolution is therefore achieved. However, the device, which operates in full depletion, requires double-sided processing of a high resistivity p-type wafer. To be processed in a standard clean room, wafers need to be about 200 μm thick for 4” wafers, and even thicker for 6” wafers. Such thicknesses, which will be entirely active as the device is fully depleted during operation, will spoil the timing capability of the LGAD. Timing resolution of only about 60 ps can be obtained for a 200 μm thick wafer. The i-LGAD can be used for imaging of particles that need some gain to be detected, for example soft X-ray, and only when timing information is not crucial. In the following, we will finally consider sensors that truly deliver 100% fill factor, fast timing and good spatial resolution.

## 5. AC-LGAD

The capacitively coupled LGAD or AC-LGAD was the first LGAD-based sensor conceived to specifically address the issue of the dead region at the border of an LGAD pixel [26,27]. An AC-LGAD is made of a large uniform gain layer implanted below the n-implant, as in the standard LGAD. Where the two devices differ is that these two layers are uniformly implanted over the entirety of the active area; there is no interruption in the layers (except at the device termination). The n-implant is covered by a thin dielectric layer of approximately 100 nm (oxide, nitride or a stack of both) which is subsequently covered with metal electrodes that are then used to connect to the read-out electronics. The n${}^{+}$ is grounded at the edge, where the JTE is present as in the case of standard LGAD, and it is surrounded by the usual termination for the high voltage handling. A sketch is shown in Figure 5. The n-implant, which can be 10 or 100 times less doped than the n${}^{+}$ implant of a standard LGAD, has a sheet resistivity much larger than the n${}^{+}$ in the conventional LGAD and can be seen as a resistive layer. The metal electrodes create coupling capacitances over the resistive n${}^{+}$ in a similar fashion as in an AC-coupled silicon sensor. The whole device can be thought of as a distributed RC network, where R is given by the n${}^{+}$ layer and C by the the capacitances towards the back and toward the metal electrodes. As the oxide is very thin, the coupling capacitance dominates over the capacitance towards the back.

The signal develops as in a standard LGAD; the electrons generated by the MIP are multiplied in the high electric field between the n${}^{+}$ and the gain layer. As the two layers are uniform, so are the electric fields and a 100% fill factor is achieved with uniform multiplication and no dead areas. The multiplied holes generate most of the signal during their drift to the p${}^{+}$ contact. The electrons are collected by the n${}^{+}$ implant and slowly drift to the DC contact at the border. The overall movement of signal electrons and holes capacitively induces a signal to the metal electrodes over the dielectric. Since the metal electrodes are separated from the n-layer by an insulator, they do not collect charge, and the signal pulse is bipolar. The fast signal “sees” a low impedance across the coupling capacitors and is fed to the read-out electronics before being spread to the neighbors. Nevertheless, a certain amount of cross talk exists due to the finite value of the interpixel resistance introduced by the n layer.

The cross talk between pixels can be exploited to increase the position resolution by interpolation at the expense of an increase of occupancy.

Tests in a laboratory setup, such as a Transient Current Technique (TCT) and confirmed at test beams, show a 100% fill factor, a timing resolution comparable to LGADs and spatial resolution at a level of 5% of the pitch [28,29]. As previously mentioned, some limitations occur as a result of the cross talk. On one hand, this is necessary to achieve excellent spatial resolution; on the other hand, it increases the occupancy so that the device in not useful in high event environments. In addition, experimental results suggest that the capacitance of long strips is quite high (the value of this capacitance is difficult to measure as it needs to be measured at high frequency). At the present stage of development, AC-LGADs in the shape of pixels or up to 1 cm long strips are excellent 4D detector candidates in low event rate environments. Pictures of LGADs recently fabricated at BNL are shown in Figure 6.

### 5.1. Deep-Layer AC-LGAD

DL AC-LGADs are a variation of the AC-LGADs which feature a deep and narrow gain layer [30]. Deep gain layers can be obtained by very high energy boron implantations (several MeV), but the resulting doping profile of the boron would be quite large. There is experimental evidence that deep and narrow gain layers increase the radiation hardness of the LGAD, currently limited to about a few $10^{15}$ n${}_{eq}$cm${}^{-2}$. A way to achieve this is to bury a low energy gain implant under a thickness of 2–3 μm of high resistivity p-type silicon, epitaxially grown. The low-energy implant will be very narrow, provided high temperature annealings are avoided in the subsequent processing steps. At the epitaxial external interface, then, the structure is the same as in a standard AC-LGAD (with JTE and p-spray or p-stop). The same performance of the standard AC-LGAD are expected. Unfortunately, the DL-ACLGAD requires clean room techniques (epitaxial deposition) which are not readily available in conventional research centers. An effort to fabricate the device is currently ongoing.

### 5.2. DC-Coupled Resistive Silicon Detectors

To limit the high cross talk in AC-LGAD, it has been recently proposed to replace the AC-coupled metal electrodes over the dielectric with DC-coupled n${}^{+}$ implants embedded into the uniform n-resistive layer. When performed this way, if the input impedance of the read-out electronics is low enough, the signal does not spread to many channels, rather it stays contained within neighboring channels. The concept is very similar to the resistive plate chamber, which uses gases as ionization and multiplication medium, where electrons generated by the MIP undergo multiplication as in an LGAD. The solution is attractive as it preserves a 100% fill factor with good timing and excellent spatial resolution as in the AC-coupled LGADs. Fabrication is in progress at FBK [31].

## 6. Deep-Junction LGAD

Another device that gives good 4D reconstruction is the deep-junction LGAD (DJ-LGAD) [32], which is currently under development. A sketch of a section of the device is shown in Figure 7. Its timing resolution is expected to be at the same level of an LGAD, while its spatial resolution is expected to be at the same level as a conventional strip or pixel detector. Therefore, it will not have the excellent spatial resolution of the AC-LGAD, but it can be used even in high rate of event environments. A deep junction, made by a large area of uniform n${}^{+}$ and p${}^{+}$ gain implants, sits a few microns from the surface where n${}^{+}$ DC coupled electrodes are placed. In this new design, the multiplication region (where high electric fields are present) is contained between the buried n${}^{+}$ and p${}^{+}$ gain layer, while everywhere else the electric fields are well below the threshold for impact ionization. Electrons multiplied in this region then drift to be collected by the n${}^{+}$ electrodes at the interface; holes, as in any type of LGADs considered so far, drift toward the uniform back. Signal electrons generated by the MIPs in the few microns between the n${}^{+}$ electrodes and the n-gain layer do not cross the high field region and therefore do not get multiplied. On the other hand, signal holes generated here cross the high field region but contribute little to the signal, as they do not undergo significant multiplication. TCAD simulations suggest that the electric field within the gain layers just below the gap between pixels is slightly lower than below the pixels, but the effect can be mitigated by a proper design of the DC electrodes. The DJ-LGAD can be fabricated by implanting both p${}^{+}$ and n${}^{+}$ gain layers on a 50-μm thick active layer and then by burying them under a few micron thickness of epitaxial layer. On the top of the layer, the DC coupled electrodes are finally defined. Another way is to implant the two gain layers on two different wafers which are then wafer-bonded together. The wafer carrying the n${}^{+}$ gain layer is then thinned down to a few microns and its surface processed to allocate the DC coupled electrodes. As the electric fields are low everywhere in the device except within the gain region, the JTE at the pixel border is no longer needed; p-spray or p-stop are instead still needed to isolate the n${}^{+}$ electrodes.

## 7. Discussion

Adding the time as an extra coordinate to resolve tracks in HEP experiments is motivating the intense research on LGAD-based devices that can deliver accurate timing information, while at least keeping the spatial resolution of the standard silicon sensors currently used. LGADs, in fact, although relatively simple to fabricate in a standard clean room process, provide poor spatial resolution. The first device developed to tackle this issue was the AC-LGAD, whose process is equally compatible with standard equipment available in a small-scale research clean room. They are very attractive as they give the best spatial resolution by interpolating the signals from a few channels which are generated by an unavoidable cross talk. The cross talk is also its main limitation for its deployment in a high event rate environment. DC-coupled resistive LGADs, recently proposed, are a variation of AC-LGADs that should have cross talk limited to the neighboring channels. Other devices require clean room techniques not readily available: TI-LGADs require trench etching, and DJ-LGADs require epitaxial layer deposition. All 4D detectors feature a large area of a uniform gain layer(s) for a 100% fill factor. The exception is the TI-LGAD, which cannot reach a 100% fill factor and benefit from deep implants, resulting in low electric fields for the same multiplication level. A summary of all the devices is reported in Table 1, with the main properties.

The silicon sensor itself is only part of the detector: fast read-out electronics are needed as well. Tests to characterize the intrinsic timing and spatial resolution properties of the silicon sensors have been carried out with single or few-channel boards mounting fast electronics based on RF or trans-impedance amplifiers, both in laboratories and at test beams.

The power budget for these boards is at the level of a few tens of mA and cannot clearly be scaled up for too many channels. For example, ATLAS [33] and CMS [34] timing detectors will use dedicated fast pixelated electronics, whose power consumption is about 1 mW/channel, and there continue to be on-going efforts which are not in the scope of the present work. The logical path forward is to integrate the read-out electronics in the same sensor substrate, using CMOS processes, as already happens with MAPS. The use of CMOS technologies brings obvious advantages (large scale fabrications, high yield, lower cost, use of state-of-the art foundries), and some efforts have already been made in that direction [25].

## Figures and Tables

**Figure 1 sensors-23-02132-f001:**
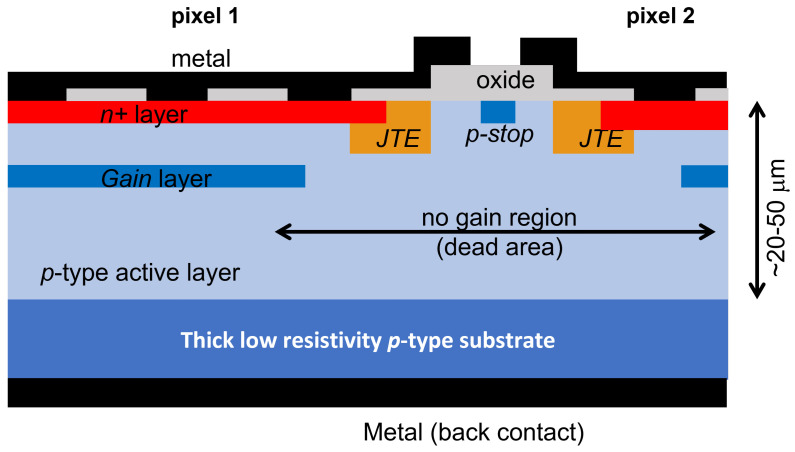
Sketch, not to scale, of a typical cross section of an LGAD, showing the main components as described in the text.

**Figure 2 sensors-23-02132-f002:**
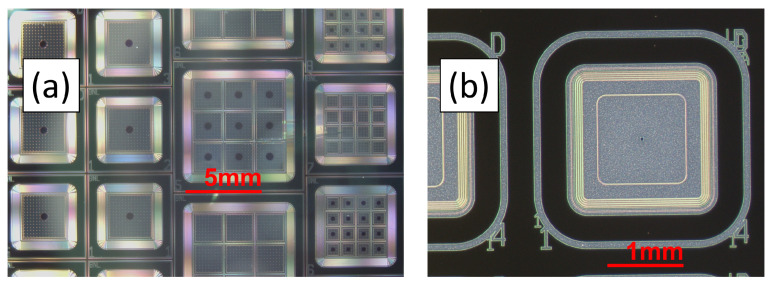
Examples of LGADs fabricated at BNL: (**a**) picture of the first production showing single channel devices and arrays; (**b**) picture of a recent production featuring 1.3 mm × 1.3 mm devices only to match the pixel size of the CMS and ATLAS timing detectors.

**Figure 3 sensors-23-02132-f003:**
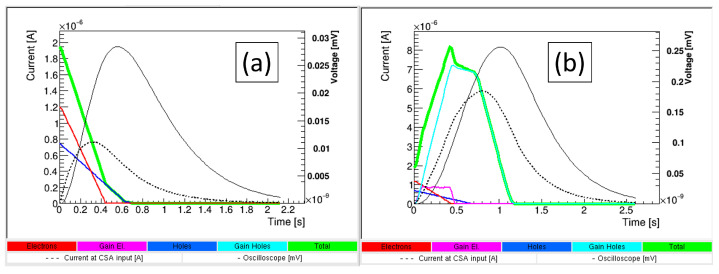
Weightfield2 [21] simulations of current pulses generated by an MIP traversing a 50-μm thick (**a**) diode and (**b**) LGAD with a gain of 10. In addition, the signal as read-out by a 500 MHz scope is shown. Red curves are current induced by primary electrons, magenta by multiplied electrons, blue by primary holes and light blue by multiplied holes. Total currents are in green.

**Figure 4 sensors-23-02132-f004:**
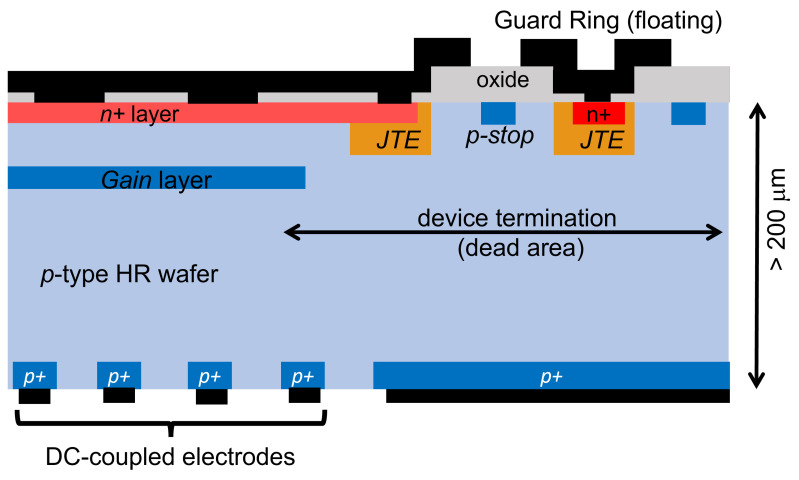
Sketch, not to scale, of a typical cross section of an i-LGAD, showing the main components as described in the text.

**Figure 5 sensors-23-02132-f005:**
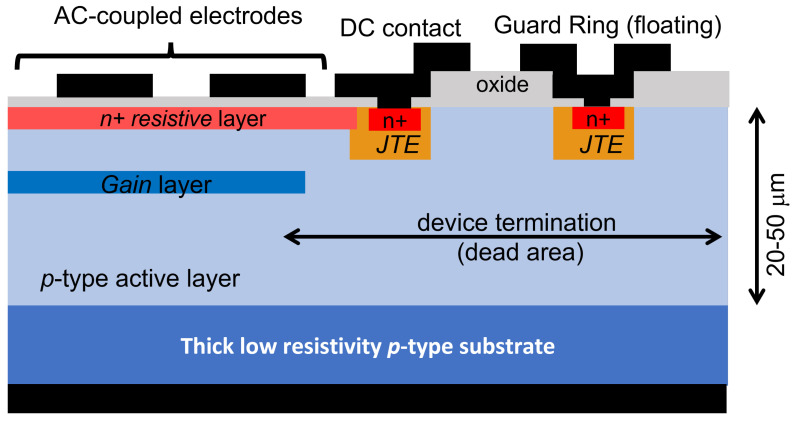
Sketch, not to scale, of a typical cross section of an AC–LGAD, showing the main components as described in the text.

**Figure 6 sensors-23-02132-f006:**
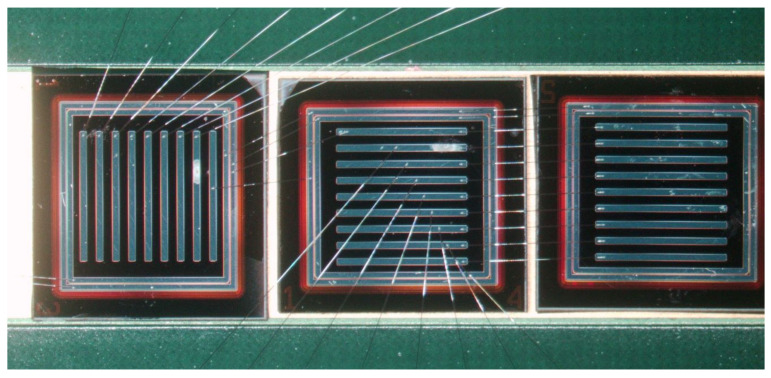
Picture of three 0.5 cm × 0.5 cm AC-LGAD devices, whose AC-coupled electrodes are in the shape of 0.5 cm long strips. Aluminum wire-bonds connect the metal electrodes to the read-out electronics (not shown) for testing purposes.

**Figure 7 sensors-23-02132-f007:**
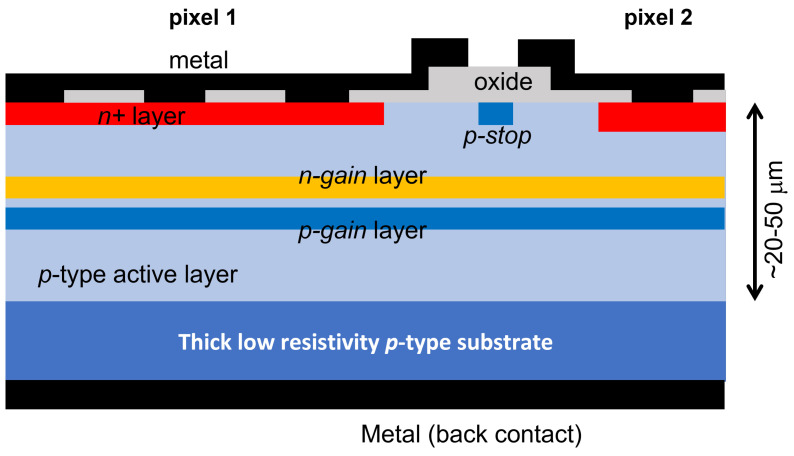
Sketch, not to scale, of a typical cross section of an DJ–LGAD, showing the main components as described in the text.

**Table 1 sensors-23-02132-t001:** Table listing all the LGAD-based devices cited in the text, with main properties. In this example, pitch is taken to be 100 μm (except for LGADs) and thickness 50 μm (except for i-LGADs).

Device	Spatial Resolution	Timing	CR Process
LGAD	∼mm	∼30 ps	std
TI-LGAD	∼10s μm	∼30 ps	trenches
i-LGAD	∼10s μm	∼60 ps	std
AC-LGAD	< 10 μm	∼30 ps	std
DL-LGAD ^1^	< 10 μm	∼30 ps	epi layer
DC RSD ^1^	∼10s μm	∼30 ps	std
DJ-LGAD ^1^	∼10s μm	∼30 ps	epi layer

^1^ For DC RSD and DJ-LGAD, spatial and timing resolutions need to be confirmed when actual devices are fabricated and tested.

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
