# Peer review of "LGAD-Based Silicon Sensors for 4D Detectors"

_sensors, 2023, doi:10.3390/s23042132_

Round 1

Reviewer 1 Report

The paper presents a review of different versions of Low Gain Avalanche Diodes presently under development, specifying in details their constructive features as well as the expected performance of each of them, in terms of spatial and time resolution. The goal of these new sensors is to provide a 4D output, which adds to the good spatial resolutions typical of a silicon-based device also a fast timing response. This will be beneficial to future High Energy Physics experiments at colliders in environments with high
luminosities, high frequency bunch crossing and very dense occupancies.
The paper is clearly written and complete, and is valuable as a general reference delivering a thorough overview of this new type of sensors. As such, it can be published in its present version once a few trivial
corrections (mainly cosmetic) are amended, as listed in the following list:

- page 2 line 8: property -> feature
- p. 2 par. 2 line 4: typical is -> typical in
- p. 3 line 2: short -> short-circuit (is "short" only jargon?)
- p. 4 line 12: dry polishing) In -> . missing: dry polishing). In
- p. 4 line 30: insert a space between the number and unit: 1 $\mu$m, 2 $\mu$m. This occurs throughout the  paper in several other places (several occurrences before "ps", see below), please review thoroughly.
- p. 4 line 36: 2e12 -> use superscript (as done somewhere else): 2$\times 10^{12}$
- p. 5 fig. 3: the labels on the colored bands specifying the color legend are too small to be read. Please report the color code in the figure caption.
- p. 6 par. 3 line 19: the acronym SPAD was never introduced before, please explain
- p. 7 par. 4 line 3: a verb is missing, probably: while the hole collecting electrodes ARE on the opposite...
- p. 7 par. 4 line 14: 60ps -> 60 ps (one of a few instances)
- p. 7 par. 5 line 10: . is missing: Figure 5. The...
- p. 9 par. 5.1 line 2: "very ion energy boron" -> not sure I understand. Do you mean "very high energy..."?
- p. 11 line 12: space missing: Table 1
- p. 11 line 5 after Table 1: amlifier -> amplifier
- p. 11 line 6 after Table 1: space missing: ATLAS [32]

Author Response

Dear Reviewer,

thank you very much for the time spent to write this detailed review.

Thank you also very much for pointing out these typos. All have been fixed.

I will also have an English speaker colleague to go one more time through the text.

- page 2 line 8: property -> feature   DONE
- p. 2 par. 2 line 4: typical is -> typical in  DONE
- p. 3 line 2: short -> short-circuit (is "short" only jargon?)   DONE
- p. 4 line 12: dry polishing) In -> . missing: dry polishing). In   DONE
- p. 4 line 30: insert a space between the number and unit: 1 $\mu$m, 2 $\mu$m. This occurs throughout the  paper in several other places (several occurrences before "ps", see below), please review thoroughly. DONE
- p. 4 line 36: 2e12 -> use superscript (as done somewhere else): 2$\times 10^{12}$ DONE
- p. 5 fig. 3: the labels on the colored bands specifying the color legend are too small to be read. Please report the color code in the figure caption.  DONE
- p. 6 par. 3 line 19: the acronym SPAD was never introduced before, please explain  DONE
- p. 7 par. 4 line 3: a verb is missing, probably: while the hole collecting electrodes ARE on the opposite... yes, the verb “are” was missing. DONE
- p. 7 par. 4 line 14: 60ps -> 60 ps (one of a few instances) DONE
- p. 7 par. 5 line 10: . is missing: Figure 5. The... DONE
- p. 9 par. 5.1 line 2: "very ion energy boron" -> not sure I understand. Do you mean "very high energy..."? correct, high energy implant. DONE
- p. 11 line 12: space missing: Table 1  DONE
- p. 11 line 5 after Table 1: amlifier -> amplifier  DONE
- p. 11 line 6 after Table 1: space missing: ATLAS [32]  DONE

Best regards

Reviewer 2 Report

The paper provides a complete overview of the LGAD-based silicon sensors for 4D detectors.

The manuscript is well organised and written. It presents a proper introduction, a well comprehensive description  of the basic unit of an LGAD sensor with relevant features as the spatial resolution and the timing, and then introduce the concepts of different types of LGAD sensor designs with the main properties.

The paper is completed with a  comprehensive summary  discussion.

The paper can be accepted for publication after only a very minor revision based on the suggestion provided in the attached paper.

However, it would interesting have also some information on the radiation hardness of such technology.

Author Response

Dear reviewer,

I’d like to thank you for the time spent in reviewing the manuscript.

A colleague of mine whose mother tongue is English went through the manuscript and performed some correction.

-I have corrected all minor points

-on Figure 3: I have extended the caption to explain the different curves in the figure (this was requested by reviewer 1). However, I cannot change the colors and the styles of the plots as the figure is generated by Weighfield2, which does not allow to export data. While it would be helpful to provide a print-friendly version of all figures, printing is not usually done nowadays so I think the Figure can stay as it is, also to credit the developers of the Weighfield2 software.

- I have updated Table 1 with numerical values.

About the lack of discussion on radiation hardness, as it is not my direct expertise, I’d like other reviews to covers the subjects.

Best  Regards